# Improving Variational Autoencoders with Density Gap-based Regularization

**Jianfei Zhang**[1,2]    **Jun Bai**[1,2]    **Chenghua Lin**[3]    **Yanmeng Wang**[4]    **Wenge Rong**[1,2]

[1]State Key Laboratory of Software Development Environment, Beihang University, China
[2]School of Computer Science and Engineering, Beihang University, China
[3]Department of Computer Science, University of Sheffield, United Kingdom
[4]Ping An Technology, China
{zhangjf,ba1_jun,w.rong}@buaa.edu.cn
c.lin@sheffield.ac.uk, wangyanmeng219@pingan.com.cn

## Abstract

Variational autoencoders (VAEs) are one of the most powerful unsupervised learning frameworks in NLP for latent representation learning and latent-directed generation. The classic optimization goal of VAEs is to maximize the Evidence Lower Bound (ELBo), which consists of a conditional likelihood for generation and a negative Kullback-Leibler (KL) divergence for regularization. In practice, optimizing ELBo often leads the posterior distribution of all samples converging to the same degenerated local optimum, namely *posterior collapse* or *KL vanishing*. There are effective ways proposed to prevent posterior collapse in VAEs, but we observe that they in essence make trade-offs between posterior collapse and the *hole problem*, i.e., the mismatch between the aggregated posterior distribution and the prior distribution. To this end, we introduce new training objectives to tackle both problems through a novel regularization based on the probabilistic *density gap* between the aggregated posterior distribution and the prior distribution. Through experiments on language modeling, latent space visualization, and interpolation, we show that our proposed method can solve both problems effectively and thus outperforms the existing methods in latent-directed generation. To the best of our knowledge, we are the first to jointly solve the hole problem and posterior collapse.

## 1   Introduction

As one of the most powerful likelihood-based generative models, variational autoencoders (VAEs) [21, 32] are designed for probabilistic modeling directed by continuous latent variables, which are successfully applied in many NLP tasks, e.g., dialogue generation [45, 14], machine translation [34, 12], recommendation [10], and data augmentation [43, 39]. One of the major advantages of VAEs is the flexible latent representation space, which enables easy manipulation of high-level semantics on corresponding representations, e.g., guided sentence generation with interpretable vector operators.

Despite the attractive theoretical strengths, VAEs are observed to suffer from a well-known problem named *posterior collapse* or *KL vanishing* [21, 24], an optimum state of VAEs when the posterior distribution contains little information about the corresponding datapoint, which is particularly obvious when strong auto-regressive decoders are implemented [46, 4].

Another challenge for VAEs is the *hole problem*, the state when the aggregated (approximate) posterior fails to fit the prior distribution, and thus the inference from the prior distribution becomes no longer suitable to describe the global data distribution [33], which can lead to poor generation quality in VAEs [1, 25].

36th Conference on Neural Information Processing Systems (NeurIPS 2022).

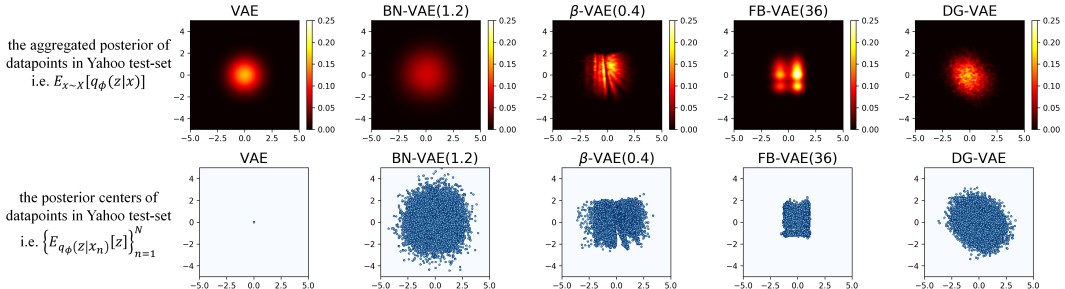

Figure 1: The visualization of the aggregated posterior distributions (the first line) and the posterior centers distributions (the second line) for models on the Yahoo test-set. The vanilla VAEs suffer from *posterior collapse*, i.e., the posterior centers collapse to the same position. Meanwhile, BN-VAEs [46], $\beta$-VAEs [18] and FB-VAEs [20] can solve posterior collapse effectively at the cost of bringing the *hole problem*, i.e., mismatch between the aggregated posterior and the prior. Our proposed DG-VAE intends to solve both problems through a novel regularization based on the *density gap*. Illustrations for more datasets, more models, and more dimensions, are shown in Appendix G.

In this work, we perform systematic experiments on VAEs for text generation to study *posterior collapse* and the *hole problem* in existing methods. We demonstrate that VAEs with specific network structures [9, 45] or modified training strategies [6, 13] have limited effect on solving posterior collapse, while VAEs with hard restrictions [8, 41, 46] or weakened KL regularization [18, 20] on the posterior distribution can solve posterior collapse effectively at the expense of the hole problem, as illustrated in Figure 1.

On that basis, we hypothesize that these two problems stem from the conflict between the KL regularization in ELBo and the function definition of the prior distribution. As such, we propose a novel regularization to substitute the KL regularization in ELBo for VAEs, which is based on the *density gap* between the aggregated posterior distribution and the prior distribution. We provide theoretical proof that our method in essence maximizes the ELBo as well as the mutual information between the input and the latent variable.

In terms of Gaussian distribution-based VAEs, we further propose the corresponding marginal regularization on each dimension respectively, and we prove it in essence maximizes the ELBo as well as the sum of mutual information between the input and the latent variable on all dimensions.

To validate our methods in practice, we take experiments on language modeling, latent-guided generation and latent space visualization. We demonstrate that our methods form latent spaces that are both active and consistent with the prior, and thus generate smoother sentences from latent interpolation. The code and data are available at `https://github.com/zhangjf-nlp/DG-VAEs`.

## 2 Background and Related Work

### 2.1 VAEs and ELBo

VAEs are proposed to perform efficient inference and learning in directed probabilistic models [21], where the random generation process consists of two steps: (1) sample a latent value $\mathbf{z}$ from the prior $p_{\boldsymbol{\theta}}(\mathbf{z})$; (2) generate a datapoint $\mathbf{x}$ from the conditional distribution $p_{\boldsymbol{\theta}}(\mathbf{x}|\mathbf{z})$. As the true posterior $p_{\boldsymbol{\theta}}(\mathbf{z}|\mathbf{x})$ is intractable, a recognition model $q_{\boldsymbol{\phi}}(\mathbf{z}|\mathbf{x})$ is introduced to approximate the true posterior [21].

Specifically, VAEs represent an observation $\mathbf{x}$ as a latent distribution $q_{\boldsymbol{\phi}}(\mathbf{z}|\mathbf{x})$, from which latent variables are sampled to direct the reconstruction of $\mathbf{x}$. As the optimization goal of VAEs, the Evidence Lower Bound (ELBo) is composed of a Kullback-Leibler (KL) divergence for regularization on the posterior and a log likelihood for reconstruction conditioned on posterior. The ELBo in fact forms a lower bound on the marginal likelihood given prior latent variables [21], as illustrated in Eq. 1.

$$\mathcal{L}_{ELBo}(\boldsymbol{\theta}, \boldsymbol{\phi}; \mathbf{x}) = \mathbb{E}_{q_{\boldsymbol{\phi}}(\mathbf{z}|\mathbf{x})}[\log p_{\boldsymbol{\theta}}(\mathbf{x}|\mathbf{z})] - D_{KL}(q_{\boldsymbol{\phi}}(\mathbf{z}|\mathbf{x})\|p_{\boldsymbol{\theta}}(\mathbf{z}))$$

$$\leq \log p_{\boldsymbol{\theta}}(\mathbf{x}) = \log \int p_{\boldsymbol{\theta}}(\mathbf{x}|\mathbf{z})p_{\boldsymbol{\theta}}(\mathbf{z})d\mathbf{z} \tag{1}$$

## 2.2 Posterior Collapse

Intuitively, the KL divergence in ELBo (i.e., $D_{KL}(q_\phi(\mathbf{z}|\mathbf{x})\|p_\theta(\mathbf{z}))$) encourages the approximate posterior distribution of every single datapoint to be close to the the prior [21]. This intends to ensure the prior distribution can depict the latent variable distribution over the data distribution, but it can also lead to posterior collapse when $D_{KL}(q_\phi(\mathbf{z}|\mathbf{x})\|p_\theta(\mathbf{z}))$ has much stronger force on $q_\phi(\mathbf{z}|\mathbf{x})$ than $\mathbb{E}_{q_\phi(\mathbf{z}|\mathbf{x})}[\log p_\theta(\mathbf{x}|\mathbf{z})]$ does, which leads that $q_\phi(\mathbf{z}|x) \approx p_\theta(\mathbf{z}) \forall x$. In such condition, the sampled latent variable, $\mathbf{z} \sim q_\phi(\mathbf{z}|x)$, contains much more noise than useful information about $x$ [27], and thus the decoder $p_\theta(\mathbf{x}|\mathbf{z})$ becomes insensitive to $\mathbf{z}$ [46].

Early works to solve posterior collapse attribute it to the difficulty in optimizing ELBo, so their methods mainly focus on the training strategies [6, 13, 23]. Some works put emphasis on the semantic learning of the latent variable through specific model structures [9, 45, 29, 17]. Instead of a Gaussian distribution, vMF-VAE [8, 41] adopts the von Mises-Fisher distribution for latent variables, which restricts the posterior latent to a hyperspherical space and forms a constant KL divergence. BN-VAE [46] restricts the posterior distribution through a batch normalization layer with a fixed scale $\gamma$, so as to guarantee a positive lower bound of the KL divergence. $\beta$-VAE [18] directly changes the weight (denoted as $\beta$) of the KL term inside ELBo, while free-bits [20] changes the KL term inside ELBo to a hinge loss term.

In contrast, we hypothesize posterior collapse is due to the conflict between the KL regularization in ELBo and the function definition of the prior distribution, and tackle it through replacing the KL regularization in ELBo with a novel regularization on the aggregated posterior distribution.

## 2.3 Hole Problem

The *aggregated (approximate) posterior* $q_\phi(\mathbf{z})$ refers to the expectation of the approximate posterior distribution on the data distribution, as defined in Eq. 2, where the distribution of observation $\mathbf{x}$ is represented by the discrete distribution of datapoints in the dataset, i.e., $X = \{x_n\}_{n=1}^N, q_\phi(\mathbf{n}) \equiv \frac{1}{N}$, in practice.

$$q_\phi(\mathbf{z}) = \mathbb{E}_\mathbf{x}(q_\phi(\mathbf{z}|\mathbf{x})) = \frac{1}{N}\sum_{n=1}^N q_\phi(\mathbf{z}|x_n) \tag{2}$$

Formally, the hole problem refers to the phenomenon that the aggregated posterior distribution $q_\phi(\mathbf{z})$ fails to fit the prior distribution $p_\theta(\mathbf{z})$. Inferences located in the holes (i.e., areas with mismatch between density in $q_\phi(\mathbf{z})$ and $p_\theta(\mathbf{z})$) are observed to generate images that are obscure and corrupted [1], or sentences with incorrect syntax and abnormal semantics [25].

The hole problem of VAEs for image generation is observed in several studies, commonly ascribed to the limited expressivity of the prior distribution and tackled by increasing the flexibility of the prior distribution via hierarchical priors [22], auto-regressive models [16], a mixture of encoders [37], normalizing flows [40], resampled priors [3], and energy-based models [1]. In contrast, we observe that the vanilla VAEs (with standard prior distributions) for text generation have no hole problem, but it arises when existing methods are applied to solve posterior collapse. Therefore, our work is targeted at solving posterior collapse and avoiding the hole problem at the same time, for VAEs with standard prior distributions.

## 2.4 Regularization on Aggregated Posterior

As the approximate posterior distribution is introduced to approximate the true posterior, i.e., $q_\phi(\mathbf{z}|\mathbf{x}) \approx p_\theta(\mathbf{z}|\mathbf{x})$, the aggregated posterior should be close to the prior as a result, i.e., $q_\phi(\mathbf{z}) \approx p_\theta(\mathbf{z})$. From this point of view, several works are proposed to replace KL regularization (on the posterior distribution of each datapoint separately) in ELBo with a regularization on the aggregated posterior distribution, which can be summarized as Eq. 3, where $D$ is the divergence (or discrepancy) between two distributions.

$$\mathcal{L}_D(\boldsymbol{\theta}, \boldsymbol{\phi}; \mathbf{x}) = \mathbb{E}_{q_\phi(\mathbf{z}|\mathbf{x})}[\log p_\theta(\mathbf{x}|\mathbf{z})] - D(q_\phi(\mathbf{z})\|p_\theta(\mathbf{z})) \tag{3}$$

Among them, Adversarial Auto-Encoder (AAE) [28] adopts the Generative Adversarial Network (GAN) [15] framework to regularize the aggregated posterior distribution through Jensen–Shannon

divergence $D_{JS}$; Wasserstein Auto-Encoder (WAE) [36, 2] regularizes the aggregated posterior distribution through minimizing Maximum Mean Discrepancy (MMD); Implicit VAE with Mutual Information regularization (iVAE$_{MI}$) regularizes the aggregated posterior through a dual form of KL divergence $D_{KL}$ on the basis of Implicit VAE (iVAE) [11]. These methods have the same weakness that their approximations of the divergence between two continuous distributions are depicted by merely sampling sets from the distributions, which involves noise from random sampling and can hardly be zero, even for the same distributions.

In contrast, our method approximates the divergence between two continuous distributions in the perspective of their mismatch in PDFs, which we quantify through the *density gap* that can be zero if and only if they are the same, which we describe in section 3.

We validate our method against the aforementioned methods through experiments on a synthetic dataset, the details and results of which are presented in Appendix A.

## 3 Methodology

**Density Gap-based Discrepancy** One of the straight manifestations of *holes* in latent space is the mismatch of probabilistic density between $q_\phi(\mathbf{z})$ and $p_\theta(\mathbf{z})$. We quantify this mismatch at a specific position $z$ in the latent space through $DG(\theta, \phi; z)$, which we refer to as *Density Gap*. Here we only consider $z \in \{z|q_\phi(z) > 0\}$.[1] We assume $q_\phi(z)$ and $p_\theta(z)$ are differentiable and $p_\theta(z) > 0$.

$$DG(\theta, \phi; z) = \log \frac{q_\phi(z)}{p_\theta(z)} = \log \frac{\frac{1}{N} \sum_{n=1}^{N} q_\phi(z|x_n)}{p_\theta(z)} \tag{4}$$

It can be inferred that the expectation of $DG(\theta, \phi; z)$ on $q_\phi(\mathbf{z})$ equals to the KL divergence between $q_\phi(\mathbf{z})$ and $p_\theta(\mathbf{z})$, as illustrated in Eq. 5, which is a strict divergence, i.e., $\mathbb{E}_{z \sim q_\phi(\mathbf{z})}[DG(\theta, \phi; z)] = 0$ iff $q_\phi(\mathbf{z}) = p_\theta(\mathbf{z})$.

$$\mathbb{E}_{z \sim q_\phi(\mathbf{z})}[DG(\theta, \phi; z)] = \mathbb{E}_{z \sim q_\phi(\mathbf{z})}[\log \frac{q_\phi(z)}{p_\theta(z)}] = D_{KL}(q_\phi(\mathbf{z}) \| p_\theta(\mathbf{z})) \geq 0 \tag{5}$$

So, we can approximate and optimize $D_{KL}(q_\phi(\mathbf{z}) \| p_\theta(\mathbf{z}))$ via Monte Carlo, as illustrated in Eq. 6, where $z_s \overset{idd}{\sim} q_\phi(\mathbf{z})$ denotes the $s^{th}$ random sample from the aggregated posterior distribution.

$$D_{KL}(q_\phi(\mathbf{z}) \| p_\theta(\mathbf{z})) \approx \frac{1}{S} \sum_{s=1}^{S} DG(\theta, \phi; z_s) \tag{6}$$

It should be noted that $D_{KL}(q_\phi(\mathbf{z}) \| p_\theta(\mathbf{z}))$ approximated by this is an *overall* divergence, as it considers the posterior distribution of all datapoints as a whole, instead of averaging $D_{KL}(q_\phi(\mathbf{z}|\mathbf{x}) \| p_\theta(\mathbf{z}))$ across all datapoints as ELBo does.

On that basis, we can implement $\mathcal{L}_D(\theta, \phi; \mathbf{x})$ with $D = D_{KL}$, which is equivalent to replacing the KL term in ELBo with $D_{KL}(q_\phi(\mathbf{z}) \| p_\theta(\mathbf{z}))$ approximated by Eq. 6. According to the decomposition (illustrated in Eq. 7) of the KL term in ELBo given by Hoffman et al. [19], maximizing $\mathcal{L}_{D_{KL}}(\theta, \phi; x_n)$ on the whole dataset, $X = \{x_n\}_{n=1}^{N}, q_\phi(\mathbf{n}) \equiv \frac{1}{N}$, is equivalent to maximizing ELBo as well as $\mathbb{I}_{q_\phi(\mathbf{n}, \mathbf{z})}[\mathbf{n}, \mathbf{z}]$,[2] the mutual information of $\mathbf{z}$ and $\mathbf{n}$ in their joint distribution $q_\phi(\mathbf{n}, \mathbf{z})$, as illustrated in Eq. 8.

$$\frac{1}{N} \sum_{n=1}^{N} D_{KL}(q_\phi(\mathbf{z}|x_n) \| p_\theta(\mathbf{z})) = D_{KL}(q_\phi(\mathbf{z}) \| p_\theta(\mathbf{z})) + \mathbb{I}_{q_\phi(n, \mathbf{z})}[n, \mathbf{z}]$$

$$\text{where } \mathbb{I}_{q_\phi(n, \mathbf{z})}[n, \mathbf{z}] = \mathbb{E}_{q_\phi(\mathbf{n}, \mathbf{z})}[\log \frac{q_\phi(\mathbf{n}, \mathbf{z})}{q_\phi(\mathbf{n}) q_\phi(\mathbf{z})}] \tag{7}$$

$$\text{where } q_\phi(n, \mathbf{z}) = q_\phi(n) q_\phi(\mathbf{z}|n) = \frac{1}{N} q_\phi(\mathbf{z}|x_n)$$

---

[1]Although we can have $q_\phi(z) = 0, z \in R^{Dim}$ when the latent variable follows a von Mises-Fisher (vMF) distribution, we do not need to consider such points in regularization.

[2]Posterior collapse (or KL vanishing) can be solved effectively by maximizing this mutual information term as it is a lower bound of the vanished KL divergence term in ELBo according to Eq. 7.

$$\frac{1}{N}\sum_{n=1}^{N}[\mathcal{L}_{D_{KL}}(\boldsymbol{\theta},\boldsymbol{\phi};x_n)] = \frac{1}{N}\sum_{n=1}^{N}[\mathbb{E}_{q_{\boldsymbol{\phi}}(\mathbf{z}|x_n)}[\log p_{\boldsymbol{\theta}}(x_n|\mathbf{z})]] - D_{KL}(q_{\boldsymbol{\phi}}(\mathbf{z})\|p_{\boldsymbol{\theta}}(\mathbf{z}))$$

$$= \frac{1}{N}\sum_{n=1}^{N}[\mathcal{L}_{ELBo}(\boldsymbol{\theta},\boldsymbol{\phi};x_n)] + \mathbb{I}_{q_{\boldsymbol{\phi}}(\mathbf{n},\mathbf{z})}[\mathbf{n},\mathbf{z}]$$

(8)

**Optimization on Mini-Batch**    Theoretically attractive as it is, maximizing $\mathcal{L}_{D_{KL}}(\boldsymbol{\theta},\boldsymbol{\phi};x_n)$ approximated by $DG(\boldsymbol{\theta},\boldsymbol{\phi};z_s)$ is undesirable for training VAEs on large datasets, because the probabilistic density of $q_{\boldsymbol{\phi}}(\mathbf{z})$ at $z$ needs computation across the whole dataset and is changing in every training step. In practice, training deep networks such as VAEs commonly adopts mini-batch gradient descent, where only a small subset of the dataset is used for calculating gradients and updating parameters in each iteration step.

So, a practicable way is to aggregate the posterior of datapoints inside a mini-batch $B = \{x_n\}_{n=1}^{|B|}, q_{\boldsymbol{\phi}}(\mathbf{n}) \equiv \frac{1}{|B|}$, as stated in Eq. 9, where $z_{n,m}$ is the $m^{th}$ sample from the posterior of datapoint $x_n$. Here, stratified sampling is used to ensure a steady Monte Carlo approximation,[3] and the reparameterization trick [21] is applied to ensure a differentiable output.

$$DG(\boldsymbol{\theta},\boldsymbol{\phi},B;z) = \log\frac{q_{\boldsymbol{\phi},B}(z)}{p_{\boldsymbol{\theta}}(z)} = \log\frac{\frac{1}{|B|}\sum_{n=1}^{|B|}q_{\boldsymbol{\phi}}(z|x_n)}{p_{\boldsymbol{\theta}}(z)}$$

$$D_{KL}(q_{\boldsymbol{\phi},B}(\mathbf{z})\|p_{\boldsymbol{\theta}}(\mathbf{z})) \approx \frac{1}{|B|}\sum_{n=1}^{|B|}\frac{1}{M}\sum_{m=1}^{M}DG(\boldsymbol{\theta},\boldsymbol{\phi},B;z_{n,m}), \text{where } z_{n,m} \overset{idd}{\sim} q_{\boldsymbol{\phi}}(\mathbf{z}|x_n)$$

(9)

Through this approximation, we can implement $\mathcal{L}_{D_{KL}}(\boldsymbol{\theta},\boldsymbol{\phi},B;x_n)$ that regularizes $q_{\boldsymbol{\phi},B}(\mathbf{z})$ towards $p_{\boldsymbol{\theta}}(\mathbf{z})$ for a mini-batch $B$, as stated in Eq. 10.

$$\frac{1}{|B|}\sum_{n=1}^{|B|}[\mathcal{L}_{D_{KL}}(\boldsymbol{\theta},\boldsymbol{\phi},B;x_n)] = \frac{1}{|B|}\sum_{n=1}^{|B|}[\mathbb{E}_{q_{\boldsymbol{\phi}}(\mathbf{z}|x_n)}[\log p_{\boldsymbol{\theta}}(x_n|\mathbf{z})]] - D_{KL}(q_{\boldsymbol{\phi},B}(\mathbf{z})\|p_{\boldsymbol{\theta}}(\mathbf{z}))$$

$$= \frac{1}{|B|}\sum_{n=1}^{|B|}[\mathcal{L}_{ELBo}(\boldsymbol{\theta},\boldsymbol{\phi};x_n)] + \mathbb{I}_{q_{\boldsymbol{\phi}}(\mathbf{n},\mathbf{z})}[\mathbf{n},\mathbf{z}]$$

(10)

It should be noticed that the mutual information term $\mathbb{I}_{q_{\boldsymbol{\phi}}(\mathbf{n},\mathbf{z})}[\mathbf{n},\mathbf{z}]$ is different in Eq. 8 (on the whole dataset) and Eq. 10 (on a mini-batch), because the range of discrete variable $\mathbf{n}$ is from 1 to $N$ in Eq. 8, but 1 to $|B|$ in Eq. 10. Consequently, $\mathbb{I}_{q_{\boldsymbol{\phi}}(\mathbf{n},\mathbf{z})}[\mathbf{n},\mathbf{z}]$ has an upper bound of $H(\mathbf{n}) = \log N$ in Eq. 8, but $H(\mathbf{n}) = \log|B|$ in Eq. 10. In other words, maximizing $\mathbb{I}_{q_{\boldsymbol{\phi}}(\mathbf{n},\mathbf{z})}[\mathbf{n},\mathbf{z}]$ in Eq. 8 intends to distinguish $\mathbf{z}$ of $x_n$ from that of $N-1$ other datapoints, while it is limited to $|B|-1$ other datapoints in Eq. 10.

**Marginal Regularization for More Mutual Information**    As described above, approximating and optimizing $D_{KL}(q_{\boldsymbol{\phi},B}(\mathbf{z})\|p_{\boldsymbol{\theta}}(\mathbf{z}))$ is practicable but has limited effect. Empirically, Gaussian distribution-based VAEs trained by Eq. 10 still have limited active units, which means the encoded latent variable $\mathbf{z}$ still collapses to the prior on most dimensions, where it provides little information.

To activate $\mathbf{z}$ on all dimensions, we propose to regularize $q_{\boldsymbol{\phi},B}(\mathbf{z})$ towards $p_{\boldsymbol{\theta}}(\mathbf{z})$ on each dimension respectively, i.e., regularize the marginal distribution of $q_{\boldsymbol{\phi},B}(\mathbf{z})$ on each dimension, as illustrated in Eq. 11, where $z_i \in R$ is the $i^{th}$ component of $z \in R^{Dim}$ and the corresponding probability density functions are of the marginal distributions on the $i^{th}$ dimension; and $z_{n,m,i}$ is the $i^{th}$ component of

---

[3]In other words, we sample $S = |B| \times M$ samples from $q_{\boldsymbol{\phi},B}(z)$ through sampling $M$ samples from $q_{\boldsymbol{\phi}}(\mathbf{z}|x_n)$ for each datapoint $x_n \in B$.

the $m^{th}$ sample from the posterior of datapoint $x_n$.

$$DG_{mrg}(\boldsymbol{\theta}, \boldsymbol{\phi}, B; z_i) = \log \frac{q_{\boldsymbol{\phi},B}(z_i)}{p_{\boldsymbol{\theta}}(z_i)} = \log \frac{\frac{1}{|B|}\sum_{n=1}^{|B|} q_{\boldsymbol{\phi}}(z_i|x_n)}{p_{\boldsymbol{\theta}}(z_i)}$$

$$D_{KL,mrg}(q_{\boldsymbol{\phi},B}(\mathbf{z})\|p_{\boldsymbol{\theta}}(\mathbf{z})) = \sum_{i=1}^{Dim} D_{KL}(q_{\boldsymbol{\phi},B}(\mathbf{z}_i)\|p_{\boldsymbol{\theta}}(\mathbf{z}_i)) \tag{11}$$

$$\approx \sum_{i=1}^{Dim} \frac{1}{|B|} \sum_{n=1}^{|B|} \frac{1}{M} \sum_{m=1}^{M} DG_{mrg}(\boldsymbol{\theta}, \boldsymbol{\phi}, B; z_{n,m,i})$$

In Gaussian distribution-based VAEs, the marginal distributions of $\mathbf{z}$ on different dimensions are independent, i.e., $q_{\boldsymbol{\phi}}(\mathbf{z}|x_n) = \prod_i^{Dim} q_{\boldsymbol{\phi}}(\mathbf{z}_i|x_n)$ and $p_{\boldsymbol{\theta}}(\mathbf{z}) = \prod_i^{Dim} p_{\boldsymbol{\theta}}(\mathbf{z}_i)$, so their KL divergence can be decomposed as $D_{KL}(q_{\boldsymbol{\phi}}(\mathbf{z}|x_n)\|p_{\boldsymbol{\theta}}(\mathbf{z})) = \sum_i^{Dim} D_{KL}(q_{\boldsymbol{\phi}}(\mathbf{z}_i|x_n)\|p_{\boldsymbol{\theta}}(\mathbf{z}_i))$. Thus, we can infer the decomposition of $D_{KL,mrg}(q_{\boldsymbol{\phi},B}(\mathbf{z})\|p_{\boldsymbol{\theta}}(\mathbf{z}))$ through Eq. 12.

$$D_{KL,mrg}(q_{\boldsymbol{\phi},B}(\mathbf{z})\|p_{\boldsymbol{\theta}}(\mathbf{z})) = \sum_{i=1}^{Dim} D_{KL}(q_{\boldsymbol{\phi},B}(\mathbf{z}_i)\|p_{\boldsymbol{\theta}}(\mathbf{z}_i))$$

$$= \sum_{i=1}^{Dim} [\frac{1}{|B|} \sum_{n}^{|B|} [D_{KL}(q_{\boldsymbol{\phi}}(\mathbf{z}_i|x_n)\|p_{\boldsymbol{\theta}}(\mathbf{z}_i))] - \mathbb{I}_{q_{\boldsymbol{\phi}}(\mathbf{n},\mathbf{z}_i)}[\mathbf{n},\mathbf{z}_i]] \tag{12}$$

$$= \frac{1}{|B|} \sum_{n}^{|B|} [D_{KL}(q_{\boldsymbol{\phi}}(\mathbf{z}|x_n)\|p_{\boldsymbol{\theta}}(\mathbf{z}))] - \sum_{i=1}^{Dim} [\mathbb{I}_{q_{\boldsymbol{\phi}}(\mathbf{n},\mathbf{z}_i)}[\mathbf{n},\mathbf{z}_i]]$$

So, maximizing $\mathcal{L}_{D_{KL,mrg}}(\boldsymbol{\theta}, \boldsymbol{\phi}, B; x_n)$ derived from Eq. 11 is equivalent to maximizing ELBo as well as the mutual information of $n$ and $\mathbf{z}_i$ for each dimension $i$ respectively, as stated in Eq. 13. In other words, it intends to distinguish $\mathbf{z}_i$ of $x_n$ from that of $|B|-1$ other datapoints for each dimension $i$ respectively. We refer to our models based on this Density Gap-based regularization as DG-VAEs.

$$\frac{1}{|B|} \sum_{n=1}^{|B|} [\mathcal{L}_{D_{KL,mrg}}(\boldsymbol{\theta}, \boldsymbol{\phi}, B; x_n)] = \frac{1}{|B|} \sum_{n=1}^{|B|} [\mathbb{E}_{q_{\boldsymbol{\phi}}(\mathbf{z}|x_n)}[\log p_{\boldsymbol{\theta}}(x_n|\mathbf{z})]] - D_{KL,mrg}(q_{\boldsymbol{\phi},B}(\mathbf{z})\|p_{\boldsymbol{\theta}}(\mathbf{z}))$$

$$= \frac{1}{|B|} \sum_{n=1}^{|B|} [\mathcal{L}_{ELBo}(\boldsymbol{\theta}, \boldsymbol{\phi}; x_n)] + \sum_{i=1}^{Dim} [\mathbb{I}_{q_{\boldsymbol{\phi}}(\mathbf{n},\mathbf{z}_i)}[\mathbf{n},\mathbf{z}_i]]$$

$$\tag{13}$$

**Aggregation Size for Ablation**  As discussed above, the size of mini-batch $|B|$ sets an upper bound of the mutual information term $\mathbb{I}_{q_{\boldsymbol{\phi}}(\mathbf{n},\mathbf{z})}[\mathbf{n},\mathbf{z}]$ (or $\mathbb{I}_{q_{\boldsymbol{\phi}}(\mathbf{n},\mathbf{z}_i)}[\mathbf{n},\mathbf{z}_i]$). To validate this impact, we further extend DG-VAEs through dividing the mini-batch into non-overlapping subsets $B = \bigcup_{i=1}^{C} b_i$, s.t. $b_j \cap b_i = \emptyset$ iff $i \neq j$, calculating and optimizing the KL divergence over each subset, i.e., $\frac{1}{C} \sum_{i=1}^{C} \frac{1}{|b_i|} \sum_{j=1}^{|b_i|} [\mathcal{L}_{D_{KL,mrg}}(\boldsymbol{\theta}, \boldsymbol{\phi}, b_i; x_n)]$, where $C$ denotes the number of subsets and $|b_i| = \frac{|B|}{C}$ is the size of those subsets, which we refer to as the *aggregation size* and denote as $|b|$ for simplification. It can be inferred that the DG-VAE with $|b| = 1$ is equivalent to the vanilla VAE trained by ELBo, except that it approximates the KL term through Monte Carlo instead of integration.

**Extension to von Mises-Fisher Distribution-based VAEs**  Besides the commonly used Gaussian distribution-based VAEs, we also consider von Mises-Fisher (vMF) distribution-based VAEs. As the decomposition $q_{\boldsymbol{\phi}}(\mathbf{z}) = \prod_i^{Dim} q_{\boldsymbol{\phi}}(\mathbf{z}_i)$ is not established for latent variables following vMF distributions (i.e., $\mathbf{z} \sim vMF(\boldsymbol{\mu}, \kappa)$), marginal regularization for vMF-VAEs may be not interpretable, so we only implement Eq. 10 in vMF-VAEs. We refer to those extensions as DG-vMF-VAEs.

Table 1: Statistics of sentences in the datasets

| Dataset | Train | Valid | Test | Vocab size | Length (avg $\pm$ std) |
|---|---|---|---|---|---|
| Yelp | 100,000 | 10,000 | 10,000 | 19997 | 98.01 $\pm$ 48.86 |
| Yahoo | 100,000 | 10,000 | 10,000 | 20001 | 80.76 $\pm$ 46.21 |
| Short-Yelp | 100,000 | 10,000 | 10,000 | 8411 | 10.96 $\pm$ 3.60 |
| SNLI | 100,000 | 10,000 | 10,000 | 9990 | 11.73 $\pm$ 4.33 |

## 4 Experiments

### 4.1 Experimental Setup

**Datasets** We consider four public available datasets commonly used for VAE-based language modeling tasks in our experiments: Yelp [42], Yahoo [42, 44], a downsampled version of Yelp [35] (we denote this as Short-Yelp), and a downsampled version of SNLI [5, 23]. The statistics of these datasets are illustrated in Table 1. It can be viewed that Yelp and Yahoo contain long sentences while Short-Yelp and SNLI contain short sentences.

**Baselines** We consider a wide range of VAEs for solving posterior collapse in text generation, where the hyperparameters are set according to Zhu et al. [46]:

- VAEs with modified training strategies (i.e., KL annealing): VAE with linear KL annealing in the first 10 epochs (default) [6]; VAE with linear KL annealing for 10 epochs at the start of every 20 epochs (cyclic-VAE) [13];

- VAEs with specific model structures: VAE with additional Bag-of-Words loss (bow-VAE) [45], and VAE with skip connection from the latent variable $\mathbf{z}$ to the vocabulary classifier for generation (skip-VAE) [9];

- VAEs with hard restrictions on the posterior distribution: $\delta$-VAE with the committed rate $\delta = 0.15$ [31]; BN-VAEs with the scale of BN layer $\gamma \in \{0.6, 0.7, 0.9, 1.2, 1.5, 1.8\}$ [46]; vMF-VAEs with the distribution's concentration $\kappa \in \{13, 25, 50, 100, 200\}$ [8, 41];

- VAEs with weakened KL regularization: FB-VAEs (free-bits) with the target KL $\lambda_{KL} \in \{4, 9, 16, 25, 36, 49\}$ [20]; $\beta$-VAEs with the weight of the KL term in ELBo $\beta \in \{0.0, 0.1, 0.2, 0.4, 0.8\}$ [18].

**Configurations** We completely follow Zhu et al. [46] in the models' backbone structures, data pre-processing, and training procedure, which we describe in detail in Appendix B.

### 4.2 Language Modeling

We evaluate the performance of our methods and the baselines on language modeling, where the following metrics are reported: the prior log likelihood $priorLL(\boldsymbol{\theta})$ and the posterior log likelihood $postLL(\boldsymbol{\theta}, \boldsymbol{\phi})$ for generation quality; the KL term in ELBo $KL(\boldsymbol{\phi})$, the mutual information $MI(\boldsymbol{\phi})$ of $\mathbf{z}$ and $\mathbf{n}$ and the number of active units $AU(\boldsymbol{\phi})$ [7] for posterior collapse; and the number of consistent units $CU(\boldsymbol{\phi})$ (we propose) for the hole problem. The corresponding expressions and explanations are presented in Appendix C.

We illustrate part of the results on Yahoo in Table 2 and all results on all datasets in Appendix D. It can be observed that: (1) models with modified training strategies or specific model structures can alleviate the problem of posterior collapse but has limited effect according to $MI(\boldsymbol{\phi})$ and $AU(\boldsymbol{\phi})$; (2) models with hard restrictions or weakened KL regularization on the posterior can solve posterior collapse better through harder restrictions or further weakening according to the increase of $KL(\boldsymbol{\phi})$, $MI(\boldsymbol{\phi})$, and $AU(\boldsymbol{\phi})$, but the decrease of $CU(\boldsymbol{\phi})$ indicates that their posterior latent spaces tend to be increasingly inconsistent with that of the prior; (3) in contrast, our proposed DG-VAE has similar performance to the vanilla VAE when $|b| = 1$, and with the increase of $|b|$, it can solve posterior collapse effectively and avoid the hole problem at the same time.[4]

---

[4]There's a little difference between DG-VAE ($|b| = 32$) and DG-VAE (default): DG-VAE ($|b| = 32$) ignores data batch $B$ if $|B| < 32$ while DG-VAE (default) accepts it through adapting to its batch size.

Table 2: Results of Language Modeling on Yahoo dataset. We bold up $MI(\phi) \geq 9.0$, $AU(\phi) \geq 30$, $CU(\phi) \geq 30$, the highest $priorLL(\theta)$ and $postLL(\theta, \phi)$ for the same methods.

| Models | $priorLL(\theta)$ | $postLL(\theta, \phi)$ | $KL(\phi)$ | $MI(\phi)$ | $AU(\phi)$ | $CU(\phi)$ |
|---|---|---|---|---|---|---|
| VAE (default) | -330.7 | -330.7 | 0.0 | 0.0 | 0 | **32** |
| cyclic-VAE | -329.8 | -328.9 | 1.1 | 1.0 | 2 | **31** |
| bow-VAE | -330.5 | -330.5 | 0.0 | 0.0 | 0 | **32** |
| skip-VAE | -330.1 | -325.2 | 5.0 | 4.3 | 8 | **31** |
| $\delta$-VAE(0.15) | -330.5 | -330.6 | 4.8 | 0.0 | 0 | 0 |
| BN-VAE(0.6) | **-327.6** | -321.1 | 6.6 | 5.9 | **32** | **32** |
| BN-VAE(1.2) | -330.9 | -310.1 | 26.2 | **9.2** | **32** | 0 |
| BN-VAE(1.8) | -343.5 | **-308.6** | 51.3 | **9.2** | **32** | 0 |
| FB-VAE(4) | -329.8 | -328.4 | 3.9 | 1.8 | **32** | **32** |
| FB-VAE(16) | **-325.7** | -320.8 | 16.1 | 8.5 | **32** | 8 |
| FB-VAE(49) | -344.6 | **-296.1** | 50.0 | **9.2** | **32** | 0 |
| $\beta$-VAE(0.4) | **-330.8** | -324.8 | 7.0 | 6.7 | 3 | **31** |
| $\beta$-VAE(0.2) | -338.6 | -310.3 | 30.1 | **9.2** | 22 | 25 |
| $\beta$-VAE(0.1) | -369.9 | **-289.6** | 83.7 | **9.2** | **32** | 0 |
| DG-VAE ($|b| = 1$) | -330.7 | -330.7 | 0.0 | 0.0 | 0 | **32** |
| DG-VAE ($|b| = 4$) | **-330.4** | -318.3 | 14.3 | **9.1** | 11 | **32** |
| DG-VAE ($|b| = 32$) | -355.4 | -294.1 | 65.2 | **9.1** | **32** | **32** |
| DG-VAE (default) | -358.0 | **-290.8** | 70.8 | **9.1** | **32** | **32** |

It can also be viewed that with the increase of $MI(\phi)$ or $KL(\phi)$, $postLL(\theta, \phi)$ tends to increase, while $priorLL(\theta)$ tends to decrease, as the decoder $\theta$ becomes more dependent on the encoder $\phi$. We further plot the curves of $priorLL(\theta)$ and $postLL(\theta, \phi)$ for models with different hyperparameters in Figure 2, where we can observe that DG-VAEs make better trade-offs than BN-VAEs and $\beta$-VAEs do on short datasets and perform competitively to BN-VAEs and FB-VAEs on long datasets.

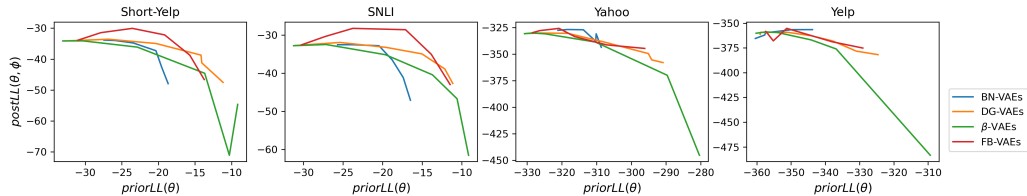

Figure 2: The curves of $priorLL(\theta)$ and $postLL(\theta, \phi)$ in Gaussian distribution-based VAEs.

We also compare the performance of DG-vMF-VAEs with vMF-VAEs under different settings of $\kappa$. As they have the same $KL(\phi)$, while $AU(\phi)$ and $CU(\phi)$ are inappropriate to report for vMF distributions, we only plot their curves of $priorLL(\theta)$ and $postLL(\theta, \phi)$ in Figure 3. It can be observed that DG-vMF-VAEs outperform vMF-VAEs in most cases.

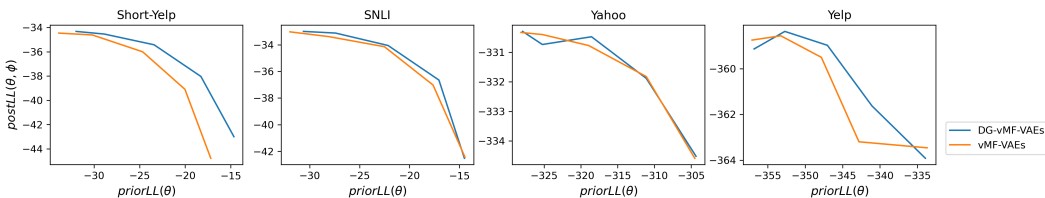

Figure 3: The curves of $priorLL(\theta)$ and $postLL(\theta, \phi)$ in vMF distribution-based VAEs.

### 4.3 Visualization of the Posterior

To further investigate the posterior distribution in latent space of those models, we visualize the aggregated posterior distributions and the posterior centers distributions on the 2 most active dimensions, i.e., the two dimensions with the highest $Var_{\mathbf{x} \sim \mathbf{X}}[\mathbb{E}_{q_\phi(\mathbf{z}|\mathbf{x})}[\mathbf{z}]]$, as depicted in Figure 4.

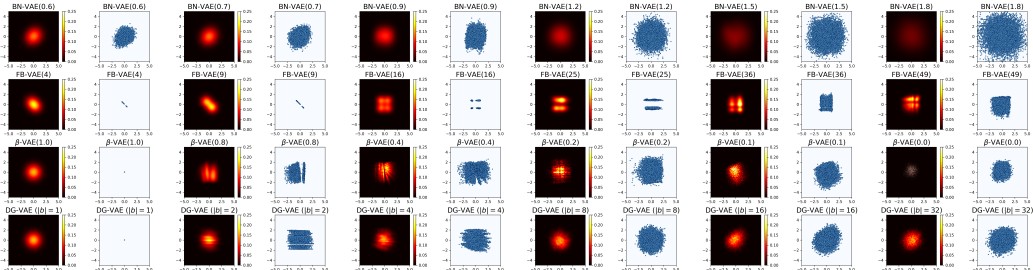

Figure 4: The visualization of the aggregated posterior distributions (red-in-black) and the posterior centers distributions (blue-in-white) for BN-VAEs, FB-VAEs, $\beta$-VAEs, and DG-VAEs on the Yahoo test-set. Illustrations for more datasets, more models, and more dimensions, are shown in Appendix G.

Here, we can observe that BN-VAEs, FB-VAEs and $\beta$-VAEs can better solve posterior collapse with harder restrictions or further weakening, but meanwhile they are faced with different kinds of mismatch between the aggregated posterior distribution and the prior distribution, i.e., the hole problem. In contrast, with the increase of aggregation size $|b|$, DG-VAE can better solve posterior collapse and avoid the hole problem in the meantime.

### 4.4 Interpolation Study

One of the main advantages of VAEs over normal language models (e.g., GPT-2 [30]) is that VAEs embed datapoints into a continuous latent space and thus enable latent-guided generation. We evaluate this ability through interpolation, where the models encode two sentences $\mathbf{x}_a$ and $\mathbf{x}_b$ as their posterior centers, i.e., $z_a = \mathbb{E}_{q_\phi(\mathbf{z}|\mathbf{x}_a)}[\mathbf{z}]$ and $z_b = \mathbb{E}_{q_\phi(\mathbf{z}|\mathbf{x}_b)}[\mathbf{z}]$, and decode the variables between them, i.e., $z_\lambda = z_a \cdot (1 - \lambda) + z_b \cdot \lambda, \lambda \in \{0.0, 0.1, \ldots, 1.0\}$.[5] The interpolated sentences are wished to be semantically smooth and meaningful, which we evaluate through the average Rouge-L F1-score [26], as stated in Eq. 14, where $F_{lcs}$ denotes the F1-score of Longest Common Subsequence (LCS). We plot the curves of Rouge-L F1-score and $\lambda$ for models on Yahoo dataset in Figure 5 and those curves on other datasets in Appendix E.

$$RougeL_{F1}(\mathbf{x}_a, \mathbf{x}_b, \mathbf{x}_\lambda) = \frac{1}{2}(F_{lcs}(\mathbf{x}_a, \mathbf{x}_\lambda) + F_{lcs}(\mathbf{x}_b, \mathbf{x}_\lambda)) \qquad (14)$$

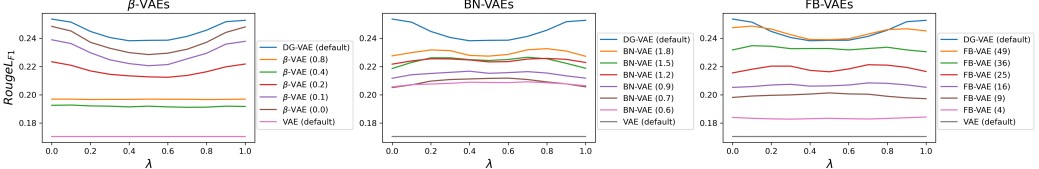

Figure 5: The curves of Rouge-L F1-score and $\lambda$ for models' interpolation performance on Yahoo.

As shown in Figure 5, the average F1-score of LCS tends to be lower in the middle than at the ends, which indicates that generated sentences tend to not be smooth in the middle, which corresponds to the phenomenon of generation near to holes observed in previous work [25][6]. The vanilla VAE performs the worst as it suffers from posterior collapse, and only generates the same plain sentence;

---

[5]We only consider greedy search for generation in this work.

[6]For further illustration on this phenomenon, we provide case study in Appendix F.

meanwhile, DG-VAE outperforms BN-VAEs, FB-VAEs and $\beta$-VAEs in the quality of interpolation on the Yahoo dataset as it can solve posterior collapse and avoid the hole problem at the same time.

In summary, the existing methods for solving posterior collapse in VAEs either have limited effect or can effectively solve posterior collapse at the cost of bringing the hole problem. In contrast, our proposed DG-VAE can effectively solve posterior collapse and avoid the hole problem at the same time, which is demonstrated by the posterior centers spread in latent space and the aggregated posterior distribution consistent with the prior distribution. Furthermore, our proposed DG-VAE outperforms the existing methods in the quality of latent-guided generation due to these improvements in latent space.

## 5   Discussion

**Conclusion**   In this work, we perform systematic experiments to demonstrate posterior collapse and the hole problem in existing continuous VAEs for text generation. To solve both problems at the same time, we propose a density gap-based regularization on the aggregated posterior distribution to replace the KL regularization in ELBo, and prove it in essence maximizes the ELBo as well as the mutual information between the latent and the input. Experiments on real-world datasets prove the effectiveness of our method in solving both problems and its improvement in latent-guided generation.

**Limitation & Future work**   Both the theory and the ablation study show that the effectiveness of our proposed method depends on the aggregation size $|b|$, which is still limited by the batch size during training. Therefore a promising future direction is to find a solution to break this limit, such like the memory bank mechanism in contrastive learning [38].

## Acknowledgments and Disclosure of Funding

This work was supported in part by the State Key Laboratory of the Software Development Environment of China under Grant SKLSDE-2021ZX-16.

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
