# OpenReview forum: "Improving Variational Autoencoders with Density Gap-based Regularization"
_NeurIPS.cc/2022/Conference — NeurIPS 2022 Accept_

### Official Review · Reviewer_rX6Q · 2022-06-25

**Rating:** 5
**Confidence:** 4
**Soundness:** 3 good
**Presentation:** 3 good
**Contribution:** 2 fair

**Summary:**

This paper proposes a unified solution to two problems which can occur when training VAEs - the hole problem (where the aggregated variational distribution fails to fit the prior) and posterior collapse (where the variational distribution becomes the same for every data point and is therefore uninformative).

The proposed solution is a modification to the VAE's training objective, where the per-data-point KL divergence term is replaced by the KL divergence from the aggregated posterior to the prior. This aggregate KL divergence term is expressed as the sum of KL divergences over the dimensions of the latent variable.

The authors compare their method to various baselines, all of which were designed to solve the posterior collapse issue. The proposed method appears to address the posterior collapse problem, while suffering the hole problem to a lesser extent than the baselines.

**Questions:**

- Is your method the first which intends to jointly solve the hole problem and posterior collapse?
    - If so, this would be a valuable statement to add.

Other questions included in the strengths and weaknesses section.

**Limitations:**

- The authors do not discuss the limitations of their method.
    - Arguably, the most important limitation of the proposed method is that the training objective is no longer a lower bound on the true log likelihood of the data. Does this mean that the model is no longer suitable for tasks such as density estimation, out-of-distribution detection, etc. which the vanilla VAE is otherwise useful for?

**Strengths And Weaknesses:**

Strengths:
- This is a conceptually simple method which appears to be effective at solving both the hole problem and posterior collapse when training VAEs.
- The authors do a good job at explaining the hole problem and posterior collapse, as well as the apparent trade-off between the two.
    - Figure 1 is particularly clear for this.
- The empirical results appear to outperform the baselines, in terms of having all of the latent dimensions being both 'active' and 'consistent' (as defined in Appendix C), as well as the latent representations and observations having a high amount of mutual information.
    - The latent space visualisations appear to indicate that the proposed method clearly outperforms the baselines at solving both the hole problem and posterior collapse.
    - In addition, the proposed method appears to consistently outperform baselines at generating interpolations between sentences.
- Although this may not be the most significant advance in generative modelling by itself, it does seem to be a piece of work that the community could easily build on in order to make incremental advances in the field.


Weaknesses:
- Although the authors explain the hole problem and posterior collapse well, the explanation of their actual method difficult is to follow. Some examples include:
    - Throughout, the use of $\mathbf{x}$ and $\mathbf{y}$ to refer to instances of the same variable is confusing. Why not just use $\mathbf{x}$ throughout as is commonly done in the literature, avoiding the introduction of unnecessary notation?
    - Although it is clear why we would expect the proposed method to solve the hole problem, it is not explained why we would expect it to solve posterior collapse? Surely, as with the vanilla VAE, there is a local optimum where $q_{\phi}(\mathbf{z}|\mathbf{x}) = p(\mathbf{z}) \forall \mathbf{x}$?
    - When reading the paper from beginning to end in order, it is unclear what the point is of Equation (3) and the paragraph just preceding it. I believe the clarity would be improved were this part moved to the part before Equation (8).
    - In Figure 1, should the $p_{\phi}$ terms not be $q_{\phi}$?
- The evaluation in the experiments section is not entirely convincing.
    - There are several missing details, e.g.
        - Are the results in Table 2 computed on the test set?
        - How many samples are used for the evaluation?
        - Why not report the (importance weighted) ELBO taken with a large number of samples, as is commonly done in the literature?
    - The interpolation task used to measure the ability of the model to do latent-guided generation isn't totally convincing.
        - The examples shown in Figures 6 and 7 are extremely short sentences, and at least qualitatively it is not clear that the DG-VAE is better than the $\beta$-VAE.
        - Why only show examples of the $\beta$-VAE and not the other baselines?

UPDATE: Increased score post rebuttal.

---

> ### Author Response · Authors · 2022-08-01
> **Response to Reviewer rX6Q**
>
> Thank you very much for your positive feedback and constructive criticism! In the following we address it point by point.
>
> > **Q:**
> 	"Throughout, the use of $\mathbf{x}$ and $\mathbf{y}$ to refer to instances of the same variable is confusing. Why not just use x throughout as is commonly done in the literature, avoiding the introduction of unnecessary notation?"
>
> **A:**
> 	We apologize for the confusion on our notations. We actually followed the notations used in [1], which used two variables $\mathbf{x}$ and $\mathbf{y}$ to represent the input and output of VAEs. Indeed, we agree that it will be a much better choice to use $\mathbf{x}$ only, which follows the standard formulation of VAEs. We have corrected the notations in the revised version of our paper.
>
> > **Q:**
> 	"Although it is clear why we would expect the proposed method to solve the hole problem, it is not explained why we would expect it to solve posterior collapse? Surely, as with the vanilla VAE, there is a local optimum where $q_{\phi}(\mathbf{z}|\mathbf{x})=p(\mathbf{z})\forall\mathbf{x}$?"
>
> **A:**
> 	Thanks for your question. The proposed objective seeks to optimize both the log-likelihood of data and the sum of marginal mutual information between the latent variable and the data. With such an objective design, the posterior collapse (or KL vanishing) can be solved effectively as the mutual information sub-objective is a lower bound of the KL divergence term in ELBo according to Hoffman et al.'s formulation.
>
> > **Q:**
> 	"When reading the paper from beginning to end in order, it is unclear what the point is of Equation (3) and the paragraph just preceding it. I believe the clarity would be improved were this part moved to the part before Equation (8)."
>
> **A:**
> 	Thanks for your valuable suggestion. We have made changes according to your suggestion in the revised version.
>
> > **Q:**
> 	"In Figure 1, should the $p_{\phi}$ terms not be $q_{\phi}$?"
>
> **A:**
> 	Thanks for pointing out the typo, which has been fixed in our revised version.
>
> > **Q:**
> 	"Are the results in Table 2 computed on the test set?"
>
> **A:**
> 	Yes, we computed the results in Table 2 on the test set following prior work [2].
>
>
> > **Q:**
> 	"How many samples are used for the evaluation?"
>
> **A:**
> 	We used the full test set for evaluation. The numbers of samples of different datasets are given in Table 1.
>
>
> > **Q:**
> 	"Why not report the (importance weighted) ELBO taken with a large number of samples, as is commonly done in the literature?"
>
> **A:**
> 	We computed the prior Log-Likelihood $priorLL(\theta)$, which shares the same information with the Negative Log-Likelihood (NLL) estimated by (importance weighted) ELBO.
>
>
> > **Q:**
> 	"The examples shown in Figures 6 and 7 are extremely short sentences, and at least qualitatively it is not clear that the DG-VAE is better than the $\beta$-VAE"
>
> **A:**
> 	Thanks for your valuable suggestion. We have added cases of long sentences on each dataset in our revised version, and highlighted tokens of the longest common subsequences for clear comparisons.
>
>
> > **Q:**
> 	"Why only show examples of the $\beta$-VAE and not the other baselines?"
>
> **A:**
> 	We only compared our method with $\beta$-VAE(0.1) in the case study as it is the best overall performing baseline model according to the automatic metrics. We will add additional results from other models in the revised version to address your comment.
>
>
> > **Q:**
> 	"Is your method the first which intends to jointly solve the hole problem and posterior collapse?" & "If so, this would be a valuable statement to add."
>
> **A:**
> 	Thanks for your valuable suggestion. To the best of our knowledge, we are indeed the first to jointly solve the hole problem and posterior collapse. We have added this statement in our revised version.
>
>
> > **Q:**
> 	"Arguably, the most important limitation of the proposed method is that the training objective is no longer a lower bound on the true log-likelihood of the data. Does this mean that the model is no longer suitable for tasks such as density estimation, out-of-distribution detection, etc. which the vanilla VAE is otherwise useful for?"
>
> **A:**
> 	Thanks for your suggestions, we don't have a definite answer to the question but we do agree that these are very intersting directions for exploration.
>
>
> [1] Yu W, Wu L, Zeng Q, et al. Crossing Variational Autoencoders for Answer Retrieval[C]//Proceedings of the 58th Annual Meeting of the Association for Computational Linguistics. 2020: 5635-5641.
>
> [2] Zhu Q, Bi W, Liu X, et al. A Batch Normalized Inference Network Keeps the KL Vanishing Away[C]//Proceedings of the 58th Annual Meeting of the Association for Computational Linguistics. 2020: 2636-2649.

---

> > ### Comment · Reviewer_rX6Q · 2022-08-06
> > **Increased score**
> >
> > Thanks to the authors for addressing the points in the review; I am happy to increase my score.
> >
> > Regarding the number of samples in the evaluations, I did not mean the number of data points but rather how many samples of $\mathbf{z}$ from the variational distribution are used in order to approximate the test set log likelihoods?

---

> > > ### Author Response · Authors · 2022-08-07
> > > **Response to Reviewer rX6Q**
> > >
> > > Thanks for your comment! We approximated the test set log likelihoods with $16$ importance weighted samples of $\mathbf{z}$ from both the variational distribution and prior distribution for each datapoint $\mathbf{x}$ (i.e., $8$ samples from each distribution), which yields robust approximations empirically. Please see also the newly provided Appendix C in the revised version for the detailed sampling process. To give more details, we conducted evaluations for all models across $10$ different random seeds under this setting and reported the mean values of log likelihoods at the precision of $0.1$, where the variances are all less than $0.01$.

---

### Official Review · Reviewer_xN7d · 2022-07-11

**Rating:** 8
**Confidence:** 4
**Soundness:** 4 excellent
**Presentation:** 4 excellent
**Contribution:** 3 good

**Summary:**

This paper focuses on a well-known problem in variational auto-encoders: the learned latent representation often has "gaps" in its prior, i.e. there are latent samples with high prior PDF values that fail to generate coherent outputs.

The authors analyse the problem and propose a novel objective to fix the issue that combines the ELBO with a mutual information term. Importantly, the proposed surrogate objective is well-motivated.

**Questions:**

- is the proposed objective still a lower bound to the log-likelihood?
- l. 123: I don't understand what is the point of stating that "we only consider  z \in ..." => q(z) is Gaussian, so we can't have q(z) = 0 anyway, no?
- in the end, the objective is only a combination of ELBo + mutual information? Could you explain a little bit why is this novel? Especially, what is the difference with [1]?

[1] MIM: Mutual Information Machine (Micha Livne et al.)

**Limitations:**

nothing to report


**Strengths And Weaknesses:**

**Strengths**

- important problem in deep generative modeling
- contribution is interesting and well-motivated
- good experimental section

**Weaknesses**

- not sure to understand whether the proposed objective is still a lower bound to the log-likelihood
- weird notation, e.g. the use of X and Y variables, it should be X only, I don't understand why to introduce a second RV for the same observation. Also, q(n) is the same as p(n), no? Why a proposal distribution here? It is fixed(?)

**Missing citations**

The following citations are missing and they provide a broader view on how researchers tackle the problem in the NLP community:
- SentenceMIM: A Latent Variable Language Model (Micha Livne, Kevin Swersky, David J. Fleet)
- A Surprisingly Effective Fix for Deep Latent Variable Modeling of Text (Bohan Li, Junxian He, Graham Neubig, Taylor Berg-Kirkpatrick, Yiming Yang)
- Preventing posterior collapse in variational autoencoders for text generation via decoder regularization (Alban Petit, Caio Corro)
- Preventing Posterior Collapse with Levenshtein Variational Autoencoder (Serhii Havrylov, Ivan Titov)

---

> ### Author Response · Authors · 2022-08-01
> **Response to Reviewer xN7d**
>
> Thanks a lot for your positive review and valuable feedback. Here are our answers for your questions:
>
> > **Q:**
> 	"not sure to understand whether the proposed objective is still a lower bound to the log-likelihood" & "is the proposed objective still a lower bound to the log-likelihood?"
>
> **A:**
> 	Thanks for your question. The proposed objective is no longer a lower bound as it seeks to optimize both the log-likelihood of data and the sum of marginal mutual information between the latent variable and the data. With such an objective design, the posterior collapse (or KL vanishing) can be solved effectively as the mutual information sub-objective is a lower bound of the KL divergence term in ELBo according to Hoffman et al.'s formulation.
>
>
> > **Q:**
> 	"weird notation, e.g. the use of X and Y variables, it should be X only, I don't understand why to introduce a second RV for the same observation."
>
> **A:**
> 	We apologize for the confusion on our notations. We actually followed the notations used in [1], which used two variables $\mathbf{x}$ and $\mathbf{y}$ to represent the input and output of VAEs. Indeed, we agree that it will be a much better choice to use $\mathbf{x}$ only, which follows the standard formulation of VAEs. We have corrected the notations in the revised version of our paper.
>
>
> > **Q:**
> 	"Also, q(n) is the same as p(n), no? Why a proposal distribution here? It is fixed(?)"
>
> **A:**
> 	Thanks for pointing out the typo. We have fixed this typo by changing $p(\mathbf{x}=x_n)$ to $q_{\phi}(n)$ to in the revised version of our paper. As stated in line 64, $n$ is the index (or identity) of datapoints whose posterior distributions compose the aggregated posterior distribution, so it is fixed to the discrete uniform distribution $q_{\phi}(n) \equiv \frac{1}{N}$.
>
>
> > **Q:**
> 	"l. 123: I don't understand what is the point of stating that "we only consider z \in ..." => q(z) is Gaussian, so we can't have q(z) = 0 anyway, no?"
>
> **A:**
> 	Besides the commonly used Gaussian distribution, we also consider von Mises-Fisher (vMF) distribution and propose corresponding variants, e.g., DG-vMF-VAEs. In theory, we can have $q_{\phi}(z)=0$ in DG-vMF-VAEs, but the domain of $DG(\theta,\phi;z)$ is $\\{z|q_{\phi}(z)>0\\}$, as we only need to compute this term for samples from $q_{\phi}(\mathbf{z})$, as stated in Equation 7. That is why we stated that we only consider $z \in \\{z|q_{\phi}(z)>0\\}$.
>
>
> > **Q:**
> 	"in the end, the objective is only a combination of ELBo + mutual information? Could you explain a little bit why is this novel? Especially, what is the difference with [1]?"
>
> **A:**
> 	Our proposed model seeks to optimize both the log-likelihood of data and the sum of marginal mutual information between the latent variable and the data. The key novelties of our models are twofold: (1) in contrast to existing models such as Adversarial Autoencoder [2] whose regularizer is merely based on sampling sets, and thus is sub-optimal, our model innovatively takes the perspective of PDFs, as we discuss in line 111, which is proved to form a continuous latent space that matches the prior much better, as illustrated in Appendix A; (2) for Gaussian settings, we present a regularizer for modelling more aggressive mutual information between the latent variable and data by imposing regularisation over marginal distributions over each dimension of the latent variable, as we introduce in line 159. This intends to make full use of the latent dimensions instead of only activating part of them.
>
> According to [3], our training objective differs from that of Asymmetric MIM mainly on the following two points: (1) MIM introduces parameterized approximate priors on data and latent variables to avoid the need for unstable adversarial training and the estimation of mutual information, while our method only adopts the anchor distributions (following vanilla VAE) and replaces adversarial training (as Adversarial Autoencoder [2] does) with our Density-Gap based regularizer; (2) Asymmetric MIM maximizes the mutual information between the data distribution and the latent variable distribution, which is similar to our Equation 10, but we further extend this to the sum of mutual information between the data distribution and the marginal distributions over each dimension of the latent variable (as illustrated in our Equation 13), so as to capture richer mutual information.
>
>
> > **Q:**
> 	"The following citations are missing and they provide a broader view on how researchers tackle the problem in the NLP community:"
>
> **A:**
> 	Thanks for your valuable sharing. We have added these related work to appropriate places in our revised version.
>
>
> [1] Yu W, Wu L, Zeng Q, et al. Crossing Variational Autoencoders for Answer Retrieval[C]//ACL. 2020: 5635-5641.
>
> [2] Makhzani A, Shlens J, Jaitly N, et al. Adversarial autoencoders[J], 2015.
>
> [3] Livne M, Swersky K, Fleet D J. MIM: Mutual Information Machine[J], 2019.

---

> > ### Comment · Reviewer_xN7d · 2022-08-09
> > **Answer**
> >
> > I upgrade my grade from 7 to 8. It would be nice to update the paper to clarify these points + add the citations from my review.
> >
> > Also, I think the paper would be more clear without these two X and Y RVs, and move to the "standard" notation for VAE, as this is sufficient for the paper.

---

> > > ### Author Response · Authors · 2022-08-10
> > > **Response to Reviewer xN7d**
> > >
> > > Thank you very much for increasing the score and for the insightful feedback! We have updated the paper in the revised version to clarify all the points raised, including using the standard VAE notation and adding the citations from your review.

---

### Official Review · Reviewer_FTv2 · 2022-07-11

**Rating:** 6
**Confidence:** 4
**Soundness:** 3 good
**Presentation:** 3 good
**Contribution:** 3 good

**Summary:**

This paper addresses posterior collapse in VAE and also tries to mitigate the issue with many existing solutions for this which is the trade off with poor fit to the prior.

The authors propose a novel regularization to substitute the KL regularization in ELBo for VAEs, which is based on the density gap between the aggregated posterior and the prior. Since quantities related to aggregated posterior need for the regularizer depend on the whole dataset are expensive to compute, this paper further changes the objective to consider aggregation over minibatches only.

This regularizer maximizes the ELBO as well as the mutual information between the input and the latent variable.

For Gaussian settings, the authors present a regularizer for more aggressive mutual information between the latent variable and data by imposing a regularizer over marginal distributions over each dimension of the latent variable.

The empirical comparison is done with relevant baselines on text datasets.


**Questions:**

some notes on the presentation:
Line 53: q(z|x) instead of y, rather x and y are confusing, its the same thing
Indicator operator used for MI
Line 82
Text in math is poorly formatted


**Limitations:**

Please address the runtime of the proposed approach.

**Strengths And Weaknesses:**

The paper is sound and the baselines are well chosen covering a range of related work.

The technical contribution hinges very heavily on the paper by Hoffman et al.'s formulation, but I believe the proposed regularizer is novel.

The interpolation study shows is interesting and the proposed approach seems to be better than the baselines on the chosen metric for interpolation. However, this metric is not directly related to mutual information between latent variables and text or the diversity one can expect from the samples of the generative model.

My main concern is that the main empirical results show that baseline/competing models are as good/better than the proposed method -- especially BN-VAE. The only metric BN-VAE does slightly worse is CU, and I am unsure how to interpret this unconventional metric.

The proposed approach also is more expensive due to the quantities involved with the aggregated posterior. A comparison of runtime with respect to other approaches would be helpful.

Some related work:
VAE with a VampPrior, Tomczak et al.
Learning to Explain: An Information-Theoretic Perspective on Model Interpretation, Chen et al.

---

> ### Author Response · Authors · 2022-08-01
> **Response to Reviewer FTv2**
>
> Thank you for the detailed and insightful review. Below, we address your points individually.
>
> > **Q:**
> 	"The interpolation study shows is interesting and the proposed approach seems to be better than the baselines on the chosen metric for interpolation. However, this metric is not directly related to the mutual information between latent variables and text or the diversity one can expect from the samples of the generative model."
>
> **A:**
> 	Thanks for your comment. Empirically, we do observe that the metric for interpolation (i.e., Rouge-L F1-score) has a strong correlation with the mutual information between latent variables and text. This is intuitive as the generated sentences tend to be irrelevant to the ground truth when the mutual information is poor, hence leading to a low Rouge-L F1-score (and vice versa).
>
> > **Q:**
> 	"The only metric BN-VAE does slightly worse is CU, and I am unsure how to interpret this unconventional metric."
>
> **A:**
> 	We propose CU to quantify the severity of the hole problem, by measuring the degree of matching between the aggregated posterior distribution and the prior distribution in a dimension-wise perspective. A lower value of CU indicates a larger mismatch between these two distributions, and hence a severer hole issue.
>
> > **Q:**
> 	"The proposed approach also is more expensive due to the quantities involved with the aggregated posterior. A comparison of runtime with respect to other approaches would be helpful." & "Please address the runtime of the proposed approach."
>
> **A:**
> 	Thanks for your insightful comment! Indeed, our approach is a bit more expensive than the baseline models due to the density gap-based regularization. However, this additional computational cost is very affordable. For instance, we compared the training time of our model and the vanilla VAE based on the default setting of batch size $|B|=32$, latent dimension $Dim=32$, and the number of samplings $M=32$ for Monte Carlo approximation in Eq. 9 and Eq. 11. The averaged training time of our model (over all experimental datasets) is only $11\\%$ higher than that of the vanilla VAE. We will include run time anlaysis results of our model and the baselines in the revised version of our paper.
>
> > **Q:**
> 	"some notes on the presentation: Line 53: q(z|x) instead of y, rather x and y are confusing, its the same thing Indicator operator used for MI Line 82 Text in math is poorly formatted"
>
> **A:**
> 	We apologize for the confusion on our notations. We actually followed the notations used in [1], which used two variables $\mathbf{x}$ and $\mathbf{y}$ to represent the input and output of VAEs. Indeed, we agree that it will be a much better choice to use $\mathbf{x}$ only, which follows the standard formulation of VAEs. We have corrected the notations in the revised version of our paper.
>
> > **Q:**
> 	"Some related work: VAE with a VampPrior, Tomczak et al. Learning to Explain: An Information-Theoretic Perspective on Model Interpretation, Chen et al."
>
> **A:**
> 	Thanks for your valuable sharing. We have added these related work to appropriate places in our revised version.
>
>
> [1] Yu W, Wu L, Zeng Q, et al. Crossing Variational Autoencoders for Answer Retrieval[C]//Proceedings of the 58th Annual Meeting of the Association for Computational Linguistics. 2020: 5635-5641.

---

> > ### Comment · Reviewer_FTv2 · 2022-08-10
> > **Thanks for your response**
> >
> > Thanks to the authors for their response to my review (and apologies for my late response to the authors). I found the response helpful -- specifically, the authors' comments on runtime have given me a better idea about the practicality of the proposed approach.

---

> > > ### Author Response · Authors · 2022-08-10
> > > **Response to Reviewer FTv2**
> > >
> > > Thank you very much for the encouraging comment! We are glad that you found our reply helpful.

---

### Meta-Review · Area_Chair_bMEr · 2022-08-23

**Recommendation:** Accept
**Confidence:** Certain

**Metareview:**

The paper addresses the KL collapse of VAE models by proposing a new regularization. Reviewers generally acknowledge the novelty of the work and have the tendency of recommending acceptance.

**Award:**

No

---

### Decision · Program_Chairs · 2022-09-14

Accept